

# Efficacy and safety of veliparib combined with traditional chemotherapy for treating patients with lung cancer: a comprehensive review and meta-analysis

Guanhua Zhao[1], Enzhi Feng[1] and Yalu Liu[2]

[1] Department of respiratory, The 941st Hospital of the People's Liberation Army, Xining, China
[2] Department of hematology, Qilu Hospital (Qingdao), Cheeloo College of Medicine, Shandong University, Qingdao, China

## ABSTRACT

**Objective.** Lung cancer, originating from bronchial mucosa or lung glands, poses significant health risks due to its rising incidence and mortality. This study aimed to assess the efficacy and safety of Veliparib combined with chemotherapy versus pharmacotherapy alone for lung cancer treatment, guiding clinical approaches for this severe disease.

**Methods.** Comprehensive searches in PubMed, EMBASE, Cochrane, and Web of Science were conducted to identify randomized controlled trials (RCTs) comparing Veliparib combined with standard chemotherapy to chemotherapy alone in lung cancer treatment, up until December 28, 2022. Two reviewers meticulously selected literature based on inclusion and exclusion criteria. The Cochrane tool was used to assess the bias risk of the included studies, and meta-analysis was performed using Stata 15.0.

**Results.** Five RCTs (1,010 participants) were included. The analysis results showed that only Veliparib combinedwith chemotherapy prolonged the progression-free survival (PFS) in small cell lung cancer (SCLC) patients [HR = 0.72, 95% CI = (0.57, 0.90)]. No significant differences were observed in overall survival (OS) and objective response rate (ORR). Veliparib and combined chemotherapy caused some side effects in patients with lung cancer, including leukopenia [RR = 2.12, 95% CI = (1.27, 3.55)], neutropenia [RR = 1.51, 95% CI = (1.01, 2.26)], anemia [RR = 1.71, 95% CI = (1.07, 3.07)], and thrombocytopenia [RR = 3.33, 95% CI = (1.19, 9.30)]. For non-small cell lung cancer (NSCLC) patients, there were no statistically significant differences in PFS, OS, or ORR between the experimental and control groups [HR = 0.97, 95% CI = (0.75, 1.27)].

**Conclusion.** The strategy of combining Veliparib with chemotherapy may, to some extent, prolong the PFS in lung cancer patients. However, this benefit is not observed in OS or ORR. Additionally, there are evident adverse reactions. Due to a limited number of the included studies, additional extensive multicenter RCTs are required to validate these results. PROSPERO registration number: CRD42023411510.

Corresponding author
Yalu Liu, liuyalucrystal@163.com

## INTRODUCTION

Lung cancer is a leading global health concern. There were 2.2 million new cases in 2020, representing 11.4% of all cancer cases. This disease also resulted in approximately 1.8 million deaths, constituting 18.0% of all cancer-related mortality (*Sung et al., 2021*; *Bray et al., 2018*). Given the difficulty in early diagnosis, most patients are diagnosed at intermediate or advanced disease stages, precluding timely surgery intervention (*Royal College of Physicians, 2017*). Although there are increasing treatment options for lung cancer, their overall efficacy for advanced-stage lung cancer remains suboptimal, and the adverse effects of drugs are significant. For instance, traditional chemotherapy and localized radiation therapy can only provide temporary symptom relief, with a five-year survival rate of only 10% to 20% (*Allemani et al., 2018*). Additionally, advancements in treatments have only marginally improved lung cancer survival rates (*Bagcchi, 2017*; *Printz, 2015*).

Both radiotherapy and chemotherapy aim to target tumor cells, damaging their DNA to induce cell death (*Olinski et al., 2002*). However, the resistance of tumor cells to chemotherapy drugs is the primary reason for clinical chemotherapy failure. Thus, studying the mechanism of multidrug resistance (MDR) remains a focal point in the current field of cancer research. These include drug efflux pumps mediated by membrane transport proteins, detoxification of tumor cells facilitated by enzymes, enhanced DNA repair functions, abnormal regulation of apoptosis genes, and anti-apoptotic mechanisms driven by signaling factors (*Gómez-Miragaya et al., 2017*). The key genes and proteins in these pathways are all associated with the induction of drug-resistant tumor phenotypes.

Poly adenosine diphosphate (ADP) ribose polymerase (PARP) proteins are activated by identifying structurally damaged DNA fragments, and detecting and marking single-strand DNA damage. They bind to DNA damage sites, synthesize ADP-ribose chains, and recruit numerous scaffold proteins and DNA repair enzymes to repair single-strand damage (*Satoh & Lindahl, 1992*). When PARP function is compromised, persistent single-strand DNA damage occurs (*De Murcia et al., 1997*). As this single-strand damage accumulates, it leads to double-strand DNA breaks. These double-strand breaks are repaired through homologous recombination repair (HRR). Proteins such as breast cancer susceptibility protein-1 (BRCA1), BRCA2, and other BRCAness proteins play a crucial role in HRR (*Bryant et al., 2005*; *Farmer et al., 2005*). When the function of these proteins is impaired, leading to HRR malfunction, alternative DNA repair mechanisms are activated, often resulting in extensive genomic rearrangements, which subsequently lead to cell death.

PARP inhibitors were initially developed to complement other therapies that cause DNA damage in cancer cells, such as radiation and chemotherapy. By diminishing the ability of cancer cells to repair DNA damage, the efficacy of other treatments would be enhanced. However, in 2005, it was discovered that tumor cells with BRCA mutations were 1000 times more sensitive to PARP inhibitors than those with wild-type BRCA genes. This significant discovery propelled the clinical use of PARP inhibitors as standalone therapies. It should be noted that while PARP inhibitors are typically associated with BRCA1 or BRCA2 gene mutations, they might also be effective against other tumors. Many tumor cells, even without BRCA1/2 gene mutations, may have HRR defects due to other reasons

and could be sensitive to PARP inhibitors. This potentially broadens the application scope of PARP inhibitors.

BRCA1/2 mutations and defective homologous recombination hinder the repair of damaged double-strand DNA, inducing cell death. Consequently, PARP is crucial in DNA damage repair and cell apoptosis (*Lord & Ashworth, 2017*). Understanding these mechanisms has led to the development of PARP inhibitors. PARP inhibitors can enhance the sensitivity of tumor cell DNA to damaging agents by inhibiting DNA repair, thereby improving the efficacy of radiation therapy and platinum-based chemotherapy. For mutations associated with BRCA gene or homologous recombination repair deficiency (HRD), using these drugs as maintenance therapy after platinum-based chemotherapy has shown clear therapeutic benefits.

Pharmacogenomics delves into the correlation between an individual's genetic makeup and drug response. This field examines the influence of genetic variations on drug metabolism, efficacy, and safety. Veliparib, belonging to the poly (ADP-ribose) polymerase (PARP) inhibitor category, is utilized in conjunction with standard chemotherapy to manage specific cancers, notably ovarian and breast cancer (*Coleman et al., 2015*; *Geyer et al., 2022*).

BRCA1/2 mutations are found in approximately 14% of non-small cell lung cancer (NSCLC) patients and about 12% of small cell lung cancer (SCLC) patients (*Ji et al., 2020*). Previous clinical trials indicated that the PARP inhibitor Veliparib, in combination with concurrent chemoradiation, achieved favorable therapeutic outcomes in NSCLC patients. However, toxicities related to cell reduction were also observed (*Ji et al., 2020*). This study uses a meta-analysis to evaluate the efficacy and safety of Veliparib combined with chemotherapy compared to chemotherapy alone in treating lung cancer patients, providing evidence more in line with the requirements of evidence-based medicine.

## MATERIALS AND METHODS

### Criteria for inclusion and exclusion

This meta-analysis followed the guidelines of the Cochrane Handbook for Systematic Reviews of Interventions (available at http://training.cochrane.org/handbook) and the Preferred Reporting Items for Systematic Review and Meta-Analyses (*Liberati et al., 2009*). The study has been registered on the International Prospective Register of Systematic Reviews (PROSPERO) with the registration number: CRD42023411510.

Before the PROSPERO registration, the research question was defined; the search strategy was formulated, and the criteria for selection (inclusion/exclusion) were set. To validate the viability of the posed question and gauge the potential number of studies to be incorporated, the search strategy was executed across various databases. After registration, the references obtained were independently evaluated by two reviewers and their results were then cross-checked.

#### Study type
Randomized controlled trials.

*Study population*

(1) Age ≥18 years; (2) Patients with confirmed lung cancer through cytology or histology, and the patients were classified into NSCLC and SCLC according to pathology type; (3) Expected life span longer than 12 weeks; (4) ECOG-PS score 0-1; (5) Adequate hematological, renal, and hepatic functions to withstand chemotherapy; (6) Patients with measurable and assessable lesions.

*Interventions*

(1) The experimental group underwent treatment with Veliparib and chemotherapy, whereas the control group received placebo with chemotherapy. (2) Both cohorts completed six intervention cycles, followed by consistent maintenance treatment until tumor progression.

*Outcomes*

The primary outcome was progression-free survival (PFS). Secondary outcomes encompassed overall survival (OS), objective response rate (ORR) and the safety of combination therapy.

*Exclusion criteria*

1. Patients without targeted diseases or those with severe comorbidities; 2. Inaccessible research data. 3. Reviews, animal experiments, pathological studies, theses or conference papers.

## Search strategy

A comprehensive literature search was conducted in Cochrane Library, PubMed, Web of Science, and Embase, to collect studies on efficacy of Veliparib in treatment of SCLC and NSCLC. The search spanned from the inception of these databases to December 28, 2022. A combination of subject terms and free keywords was used to design the search strategy, such as Veliparib, NSCLC, and SCLC. No restrictions were imposed on language of literature. The specific search strategy is provided in Table S1.

## Selection of studies and extraction of data

Two researchers (Yalu Liu, Guanhua Zhao) meticulously scrutinized the collected literature, adhering to predefined inclusion and exclusion criteria. Endnote X9 was utilized to manage the retrieved articles, and duplicates were removed. Studies were initially screened based on titles or abstracts, and their full texts were downloaded. After reading the full texts, the original studies that met the criteria for this systematic review were selected. Data were extracted from the included studies and cross-checked, and units of measurement were standardized. In cases of disagreements, a decision was reached after discussion with a third researcher. The extracted information primarily included the title, first author, publication year, country, study type, pathological type, sample sizes of the experimental and control groups, number of males and females and their ages, intervention methods, intervention duration, stage of medication, follow-up duration, drug dosage, and outcome indicators.

## Assessment of bias risk in the selected studies

Two researchers independently assessed the risk of bias in the studies, and their results were cross-checked. The Cochrane risk of bias tool was used to evaluate the quality of the included studies, covering 7 aspects: generating random sequences (selection bias), concealment of allocation (selection bias), blinding of researchers and subjects (implementation bias), blinding in the evaluation of outcomes(measurement bias), completeness of outcome data (follow-up bias), selective study result presentation (reporting bias), and additional potential sources (other biases).

## Statistical methods

Statistical analysis was executed using Stata 15.0 (StataCorp LLC, College Station, TX, USA), including heterogeneity tests, sensitivity assessments, and publication bias evaluations. Hazard ratios (HRs) were transformed to "lnHR" and "selnHR", and subsequently were pooled. Their 95% confidence intervals (CIs) were provided. Dichotomous variables were expressed as relative risk (RR) with 95% CI. The Q statistic and $I^2$ test were used to quantify heterogeneity. A fixed-effects model was applied for meta-analysis if the heterogeneity among the studies was acceptable ($P>0.1$ and $I^2 \leq 50\%$), $P \leq 0.1$ or $I^2>50\%$ indicated significant heterogeneity, and thus a random-effects model was used. Publication bias in the studies was identified using the "metabias" command. A $P<0.05$ was deemed statistically significant.

# RESULTS

## Literature retrieval

The database search yielded 3029 articles. Using EndNote X9, 1064 duplicates were removed. Upon reviewing titles and abstracts, another 1965 unrelated articles were eliminated. After a thorough review of the full texts, 31 ineligible articles and two with inaccessible full texts were excluded. Thus, five articles were selected (Fig. 1).

## Fundamental characteristics of included literature

The five studies *Ramalingam et al. (2017)*, *Pietanza et al. (2018)*, *Govindan et al. (2022)*, *Argiris et al. (2021)* and *Byers et al. (2021)* enrolled a total of 1,010 participants, with 537 in the experimental group and 473 in the control group. The experimental group received Veliparib combined with chemotherapy, whereas the control group was treated with placebo combined with chemotherapy. All the included RCT studies were in English (Table 1).

## Evaluation of selected literature

Among the included articles, one study did not specify the method for random grouping. All studies used the double-blind method. It was unclear in two studies whether the baseline and subsequent measurements were conducted by the same measurer. All studies had no risk of attrition bias, reporting bias, or other bias (Fig. 2).

Zhao et al. (2023), *PeerJ*, DOI 10.7717/peerj.16402

Peer J

**Table 1** **The fundamental characteristics of the included literature.**

| Author, Year | Country | Cancer subtype | Study type | Treatment | Sample Size | | Age (year) | | Follow Up time | Outcome indicators |
|---|---|---|---|---|---|---|---|---|---|---|
| | | | | | Experiment group | Control group | Experiment group | Control group | | |
| Suresh S. Ramalingam 2017(7) | USA | NSCLC | RCT | Veliparib + Carboplatin and Paclitaxel | 105 | 52 | 63 (33–84) | 62 (46–79) | 6 cycles | PFS/OS/side effect |
| M. Catherine Pietanza 2018(8) | USA | SCLC | RCT | Veliparib + Temozolomide | 55 | 49 | 63 (31–80) | 62 (35–84) | 4 months | PFS/ORR/OS/safety and tolerability |
| Ramaswamy Govindan 2022(9) | USA | NSCLC | RCT | Veliparib + Carboplatin and Paclitaxel | 298 | 297 | 63 (27–81) | 64 (34–85) | 6 cycles | PFS/OS/ORR |
| Athanassios Argiris 2021(10) | Greece | NSCLC | RCT | Veliparib + Carboplatin and Paclitaxel | 18 | 13 | 64 (47–79) | 65 (57–76) | 6 cycles | PFS/OS/response rate/ toxicities |
| Lauren Averett Byers 2021(11) | USA | SCLC | RCT | Veliparib + Carboplatin and Etoposide | 61 | 61 | 62 (39–77) | 63 (37–87) | 4–6 cycles | PFS/OS/ORR/DOR |

**Notes.**
RCT, Randomized controlled trial.
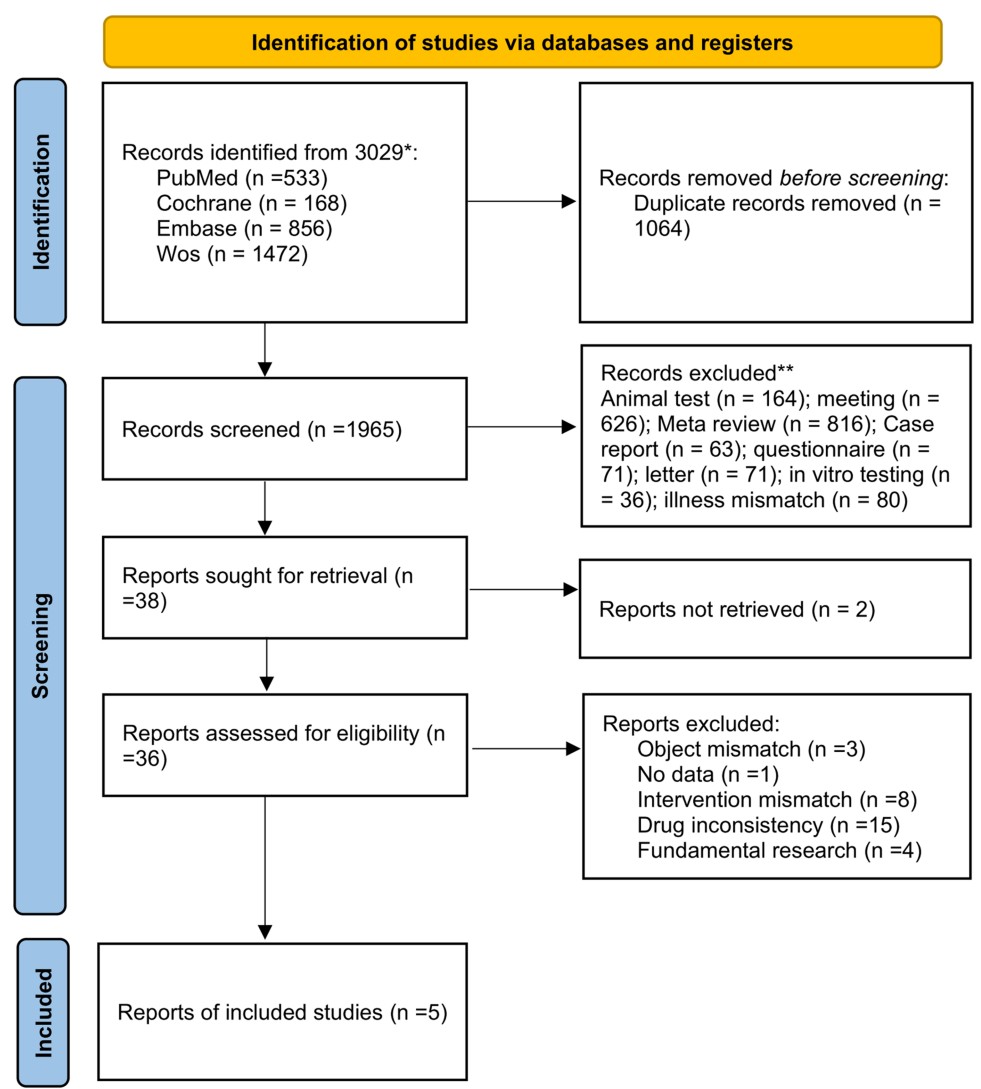

**Figure 1** **Procedure outline for the screening of literature.**

## Meta-analysis results
### PFS

Five studies reported the impact of Veliparib combined with chemotherapy on PFS in lung cancer patients. The effect sizes were pooled using a random-effects model ($I^2 = 55.5\%$, $P = 0.061$). No significant statistical difference was observed in PFS between the experimental group and the control group [HR = 0.85, 95% CI = (0.67, 1.09)]. Subgroup analysis by pathology (NSCLC and SCLC) revealed that Veliparib with chemotherapy strategy enhanced PFS in SCLC patients [HR = 0.72, 95% CI = (0.57, 0.90)], but not in NSCLC patients [HR = 0.97, 95% CI = (0.75, 1.27)] (Fig. 3).

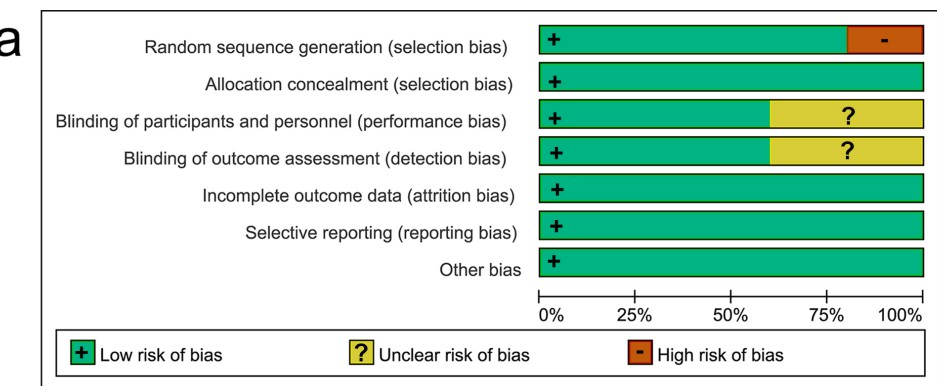

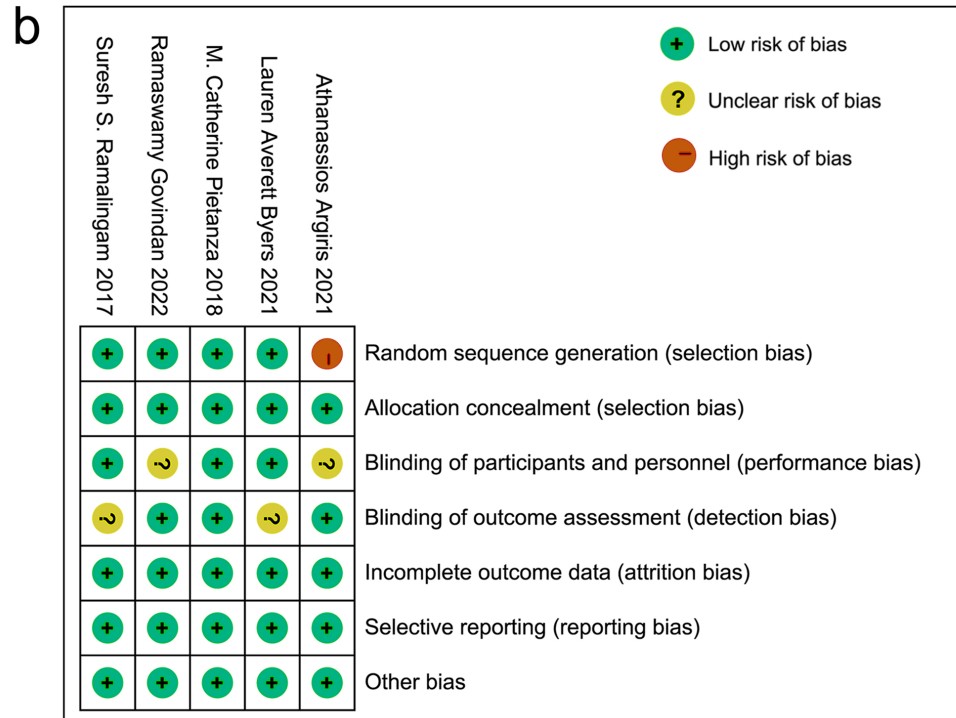

**Figure 2** **Evaluation of bias risk within the chosen literature.** (A) Graphic depiction of bias risk; (B) concise summary of bias risk.

## OS

Five studies discussed the influence of Veliparib combined with chemotherapy on OS in lung cancer patients. A random-effects model was applied for data analysis ($I^2 = 52.5\%$, $P = 0.077$). No significant statistical variation was seen in OS between the experimental group and the control group [HR = 1.02, 95% CI = (0.81, 1.29)]. Subgroup analysis by pathology (NSCLC and SCLC) indicated that Veliparib with chemotherapy had no substantial therapeutic effect in both SCLC [HR = 1.23, 95% CI = (0.83, 1.81)] and NSCLC patients [HR = 0.94, 95% CI = (0.80, 1.10)] (Fig. 4).

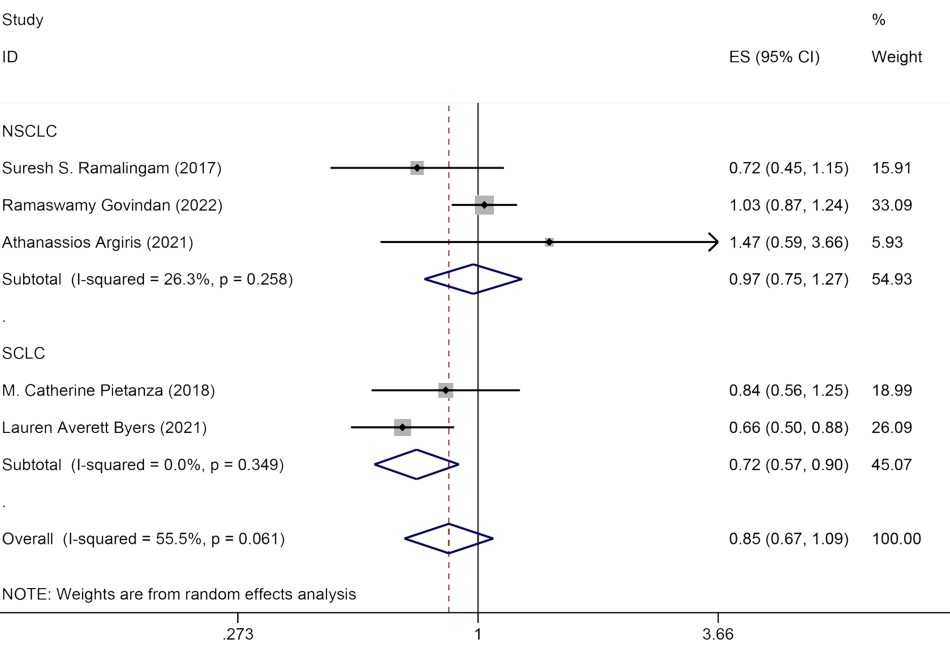

**Figure 3** Forest plot for representing the efficacy of Veliparib when combined with chemotherapy to the PFS.

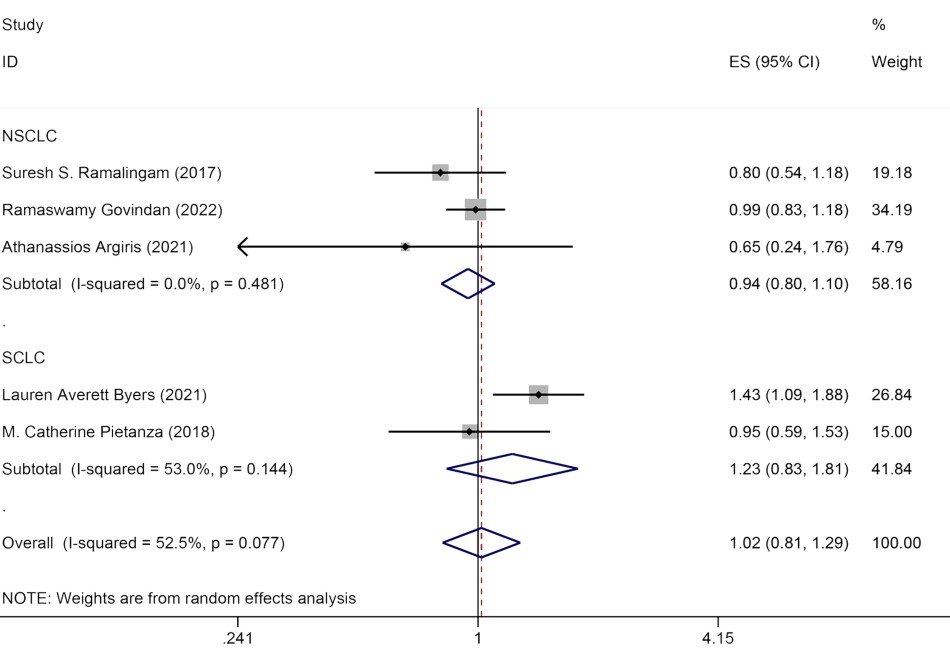

**Figure 4** Forest plot for illustrating the impact of Veliparib combined with chemotherapy on the OS.

### ORR

Five studies evaluated the impact of Veliparib in combination with chemotherapy on the ORR among lung cancer patients. A random-effects model was used for analysis ($I^2 =$

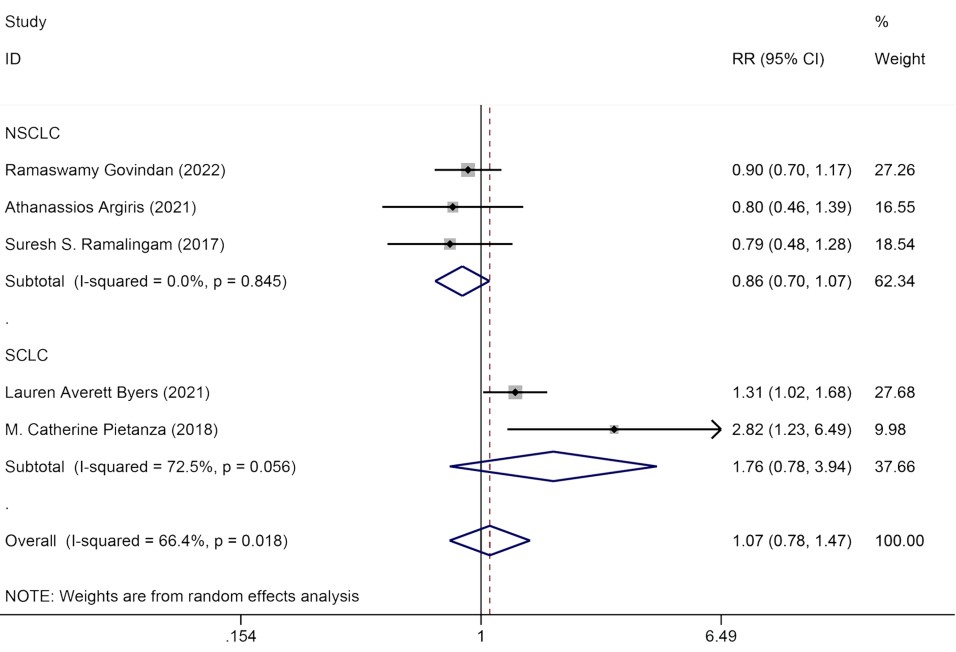

**Figure 5** Forest plot for illustrating the impact of Veliparib combined with chemotherapy on the ORR.

66.4%, $P = 0.018$). No significant statistical difference was detected in ORR between the experimental group and the control group [HR = 1.07, 95% CI = (0.78, 1.47)]. Subgroup analysis by pathology (NSCLC and SCLC) showed that Veliparib with chemotherapy had no notable therapeutic benefits for either SCLC [HR = 1.76, 95% CI = (0.78, 3.94)] or NSCLC patients [HR = 0.86, 95% CI = (0.70, 1.07)] (Fig. 5).

### Safety
Five studies reported the safety of Veliparib combined with chemotherapy. Fixed-effects models were used to pool effect sizes for leukopenia, anemia, anorexia, nausea, vomiting, and fatigue ($I^2 = 0.0\%$, $P = 0.599$; $I^2 = 0.0\%$, $P = 0.560$; $I^2 = 0.0\%$, $P = 0.545$; $I^2 = 47.9\%$, $P = 0.124$; $I^2 = 0.0\%$, $P = 0.555$; and $I^2 = 0.0\%$, $P = 0.584$). For neutropenia and thrombocytopenia, random-effects models were employed ($I^2 = 53.6\%$, $P = 0.071$ and $I^2 = 53.0\%$, $P = 0.094$, respectively). The findings indicated that Veliparib with chemotherapy significantly raised the risk of blood cell reduction, including leukopenia [RR = 2.12, 95% CI = (1.27, 3.55)], neutropenia [RR = 1.51, 95% CI = (1.01, 2.26)], anemia [RR = 1.71, 95% CI = (1.07, 3.07)], and thrombocytopenia [RR = 3.33, 95% CI = (1.19, 9.30)]. However, there were no significant differences in other major side effects like anorexia [RR = 0.86, 95% CI = (0.35, 2.10)], nausea [RR = 0.74, 95% CI = (0.36, 1.53)], vomiting [RR = 0.66, 95% CI = (0.25, 1.72)], and fatigue [RR = 2.59, 95% CI = (0.79, 8.49)] between the two groups (Fig. S1).

### Sensitivity assessment and publication bias detection
Publication bias was assessed, and no evidence of publication bias was found. Sensitivity analysis for fatigue, anemia and decreased platelets highlighted that the study by

*Govindan et al. (2022)* was highly sensitive. After excluding this study, the sensitivity analysis results returned to normal.

## DISCUSSION

The systemic anti-tumor treatment for lung cancer has undergone transformations from chemotherapy, radiation therapy, targeted therapy, to immunotherapy. Despite such progress, platinum-based chemotherapy remains the main therapy for lung cancer. A primary challenge remains drug resistance following chemotherapy, which restricts the effectiveness of cancer treatments. Consequently, enhancing the efficacy of chemotherapy and identifying effective treatment strategies after platinum-based chemoresistance have always been research hotspots in lung cancer treatment.

Cisplatin is frequently utilized in clinical practice, which may induce DNA damage during lung cancer treatment. PARP is important for DNA damage repair. In 2007, both cell and animal studies verified that the PARP inhibitor Veliparib could inhibit DNA repair and significantly suppress tumor growth (*Albert et al., 2007*). Furthermore, Veliparib, when paired with DNA alkylating agents (like temozolomide and radiotherapy), possesses potent anti-tumor properties (*Donawho et al., 2007*). In a 2012 phase I clinical trial, Veliparib exhibited excellent safety, pharmacokinetics, and pharmacodynamics, with 13 participants experiencing clinical benefits from the treatment (*Kummar et al., 2012*).

Pharmacogenomics is crucial in understanding the safety and efficacy of Veliparib combined with standard chemotherapy. Genetic variations can influence the processing of Veliparib in the body, leading to variations in drug metabolism and treatment outcomes. Examining specific genetic markers provides insights into the prediction of the likelihood of patient's responses to Veliparib and risks of potential drug-related adverse reactions. Hence, pharmacogenomics is vital for evaluating the safety and efficacy of Veliparib with standard chemotherapy. Patient's genetic constitution, especially their BRCA mutation status, can help guide healthcare providers in choosing appropriate treatment plans, potentially resulting in enhanced therapeutic outcomes and fewer side effects.

Up to now, PARP inhibitors have been extensively utilized in the oncology field for several years and have been approved by FDA for the treatment of breast, ovarian, and prostate cancers. Nonetheless, the therapeutic potential of PARP inhibitors in thoracic malignancies remains uncertain. *Mehta et al. (2015)* detailed a phase I trial on the effects of Veliparib, a PARP inhibitor, paired with whole-brain radiotherapy for brain metastasis patients. This combined treatment exhibited notable benefits in patients with brain metastases originating from NSCLC and breast cancer, with a median PFS of 10.0 and 7.7 months, respectively. Both figures surpassed the average PFS time anticipated from the Kaplan–Meier curve (*Mehta et al., 2015*). Another phase I study by *Kozono et al. (2021)* explored the combination of Veliparib and chemoradiotherapy for patients with inoperable stage III NSCLC. Veliparib combined with standard chemoradiotherapy and consolidation chemotherapy was potent in anti-tumor treatment, with a median PFS of 19.6 months. All side effects from this combination treatment were manageable, with no additional toxicities noted (*Kozono et al., 2021*).
This study assessed the efficacy and safety of Veliparib combined with chemotherapy *versus* chemotherapy alone, in the treatment of lung cancer. This meta-analysis included five clinical trials. The data revealed no significant statistical difference in PFS between the two groups. However, notable heterogeneity was observed in primary endpoints among the included studies, potentially attributed to variations in pathology type, treatment strategy, drug type, and drug administration timing.

In the subgroup analysis by pathology, Veliparib combined with chemotherapy prolonged PFS in patients with SCLC. However, no significant improvements were noted in either ORR or OS. For the NSCLC patient group, combining Veliparib with chemotherapy did not improve PFS, OS, or ORR. While Veliparib showed some anti-tumor effects in patients with SCLC, this prolonged PFS did not translate to an OS benefit. A report by Byers et al. corroborates this observation, as they highlighted that Veliparib plus platinum chemotherapy followed by Veliparib maintenance enhanced PFS as the first-line treatment for SCLC, which is consistent with the findings in this analysis (*Byers et al., 2021*).

Previous studies indicated that PARP inhibitors might induce more DNA damage, increase neoantigens, and elevate PD-L1 expression through interferon, potentially amplifying the response to immune checkpoint inhibitors (*Vikas et al., 2020*). Clinical trials have confirmed that the combination of immune-checkpoint inhibitors and PARP has potential efficacy in patients with metastatic breast, ovarian, prostate, and NSCLC cancers (*Konstantinopoulos et al., 2019*; *Karzai et al., 2018*; *Ramalingam et al., 2022*). Consequently, PARP inhibitors in combination with chemotherapy or immunotherapy may be a promising treatment option for lung cancer patients.

OS, the gold standard for evaluating the efficacy of anti-tumor drugs, is increasingly difficult to be improved in the short term, despite the introduction of new therapeutic approaches and the optimization of treatment modes. Consequently, PFS has become the most widely used surrogate endpoint in clinical research. Nevertheless, there are limitations to PFS. For instance, in clinical practice, relapse is typically diagnosed using radiology rather than pathology, and the time between relapse and follow-up can impact the assessment.

Veliparib combined with chemotherapy exhibited significant drug-induced side effects, including leukopenia, neutropenia, anemia, and thrombocytopenia. A statistically significant difference was observed in side effects between the two groups. Furthermore, Veliparib causes bone marrow suppression.

Our study is the first to provide an evidence-based basis for the treatment of NSCLC and SCLC with PARP (Veliparib) based on high-quality RCTs, furnishing important references for the selection of treatment options in subsequent clinical practice. Nonetheless, this study had several limitations. First, some of the included studies are of low quality with a limited number of cases, which may impact the research outcomes. Second, the pathology types, disease staging, treatment strategies, and timing of medication in the incorporated studies are not consistent. Third, due to the limited follow-up duration, an effective evaluation of overall survival was not feasible. Therefore, high-quality, large-sample, multi-center RCTs are needed to verify these findings.

## CONCLUSION

Veliparib combined with chemotherapy may somewhat enhance the PFS in lung cancer patients. However, this benefit is not observed in OS or ORR, and there is a significant increase in adverse reactions related to blood cell reduction. Given the limited number of included studies, further high-quality, multi-center RCTs are required to validate this conclusion.

### Funding

The authors received no funding for this work.

### Competing Interests

The authors declare there are no competing interests.

### Author Contributions

- Guanhua Zhao conceived and designed the experiments, performed the experiments, analyzed the data, prepared figures and/or tables, authored or reviewed drafts of the article, and approved the final draft.
- Enzhi Feng conceived and designed the experiments, performed the experiments, analyzed the data, prepared figures and/or tables, and approved the final draft.
- Yalu Liu conceived and designed the experiments, performed the experiments, analyzed the data, prepared figures and/or tables, authored or reviewed drafts of the article, and approved the final draft.

### Data Availability

This is a meta-analysis.

### Supplemental Information

Supplemental information for this article can be found online at http://dx.doi.org/10.7717/peerj.16402#supplemental-information.

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
