# Peer review of "Efficacy and safety of veliparib combined with traditional chemotherapy for treating patients with lung cancer: a comprehensive review and meta-analysis"

_PeerJ, doi:10.7717/peerj.16402_

## Round 0.1 · original submission · Major Revisions

Authors should revise according to the suggestions of reviewers. The modifications should be marked. A point to point response letter is needed.

**Language Note:** The review process has identified that the English language must be improved. PeerJ can provide language editing services - please contact us at [email protected] for pricing (be sure to provide your manuscript number and title). Alternatively, you should make your own arrangements to improve the language quality and provide details in your response letter. – PeerJ Staff

Reviewer 1 ·

Basic reporting

This scholarly endeavor seeks to comprehensively evaluate the effectiveness and safety profile of the synergistic amalgamation between veliparib and conventional chemotherapy in the management of lung cancer. Within this pursuit, there exist some areas of improvement that merit attention, as explicated below:

 The authors are encouraged to draw upon the reservoir of published literature, ensuring the meticulous articulation of the 95% confidence interval (95% CI) value in its accurate form.

 In the discourse section, a particular emphasis should be placed upon elucidating the innovative facets inherent within the authors' endeavor. Moreover, an elaborate discourse on the practical applicability of their discoveries within the clinical domain is warranted.

 Table 1 necessitates the incorporation of interventions, with particular attention to specifying the chemotherapy agents employed.

 The linguistic proficiency exhibited in the manuscript was found wanting, leading to instances of obscurity. Thus, it is recommended that the manuscript be subjected to a rigorous linguistic and scientific review. This can be performed by either a proficient English speaker or a professional language editing service, encompassing both the realms of English language refinement and scientific rigor.

Experimental design

None

Validity of the findings

None

Additional comments

None

Reviewer 2 ·

Basic reporting

no comment

Experimental design

no comment

Validity of the findings

The researchers have convincingly demonstrated that the synergistic combination of veliparib and chemotherapy has the potential to moderately enhance Progression-Free Survival (PFS) in the context of lung cancer patients, primarily due to its profound implications in the realm of pharmacogenomics. The study embarks on a captivating journey of exploration, delving into intricate nuances.

However, there exist certain dimensions within the study that merit thoughtful consideration:

1) It is strongly advised that the authors seamlessly incorporate the references numbered two, three, and four into the Introduction section, enriching the contextual framework.

2) Within Figure 1, kindly ensure the inclusion of both the full title and the corresponding abbreviation for "Wos," enhancing clarity and accessibility.

3) A deeper level of elucidation is warranted within the legend notes of Figure 2. Specifically, there is a need for explicit clarification regarding the interpretations behind the symbols "+," "–," and "?," enhancing the comprehensive understanding of the depicted data.

4) Regrettably, the manuscript's eloquence is marred by instances of suboptimal writing quality, characterized by a pervasive presence of typographical and grammatical errors that persist across its entirety. Addressing these issues will undoubtedly elevate the manuscript's overall impact and readability.

Additional comments

no comment

Reviewer 3 ·

Basic reporting

The researchers conducted a comprehensive systematic review and meta-analysis to assess the safety and efficacy of combining veliparib with conventional chemotherapy for the treatment of lung cancer. This investigation holds significant research implications, as it explores the potential benefits of this novel approach. The manuscript effectively delves into an intriguing and vital subject matter. However, critical issues pervade the paper, undermining its overall excellence. The authors are urged to address these concerns meticulously before contemplating submission for publication.

1. It is imperative to consistently expand all abbreviations in the manuscript, providing their complete terminology upon first usage.
2. Within the "Materials and Methods" section, specifically in the 2.2 Search Strategy, a discrepancy is apparent. While the author asserts that no language restrictions were imposed, the subsequent search neglected articles in languages other than English, notably Chinese. To rectify this, the recommendation is to revise the search parameters to exclusively encompass English language sources.
3. In the "Results" section, precisely in the 3.2 Fundamental Characteristics of Included Literature subsection, the elucidation of findings falls short in its complexity. The content housed in Table 1 necessitates more comprehensive elaboration to convey a thorough understanding.
4. The inclusion of a subgroup analysis is strongly advised, enriching the manuscript's depth and insights.
5. The quality of the manuscript's written English requires refinement. Engaging the services of a fluent English speaker is recommended to enhance linguistic precision and fluency.

Experimental design

null

Validity of the findings

null

Additional comments

null

---

## Round 0.2 · accepted · Accept

The authors have addressed the reviewers' concerns properly and revised the manuscript accordingly. The manuscript can be accepted for publication in its current form.

Reviewer 2 ·

Basic reporting

no comment'

Experimental design

no comment'

Validity of the findings

The authors have addressed all the issues we concerned.

Additional comments

no comment'

Reviewer 3 ·

Basic reporting

No report.

Experimental design

No report.

Validity of the findings

No report.

Additional comments

No report.

Annotated reviews are not available for download in order to protect the identity of reviewers who chose to remain anonymous.